# Context-Dependent Role of Glucocorticoid Receptor Alpha and Beta in Breast Cancer Cell Behaviour

**DOI:** 10.3390/cells12050784

**Published:** 2023-03-01

**Authors:** Henriett Butz, Éva Saskői, Lilla Krokker, Viktória Vereczki, Alán Alpár, István Likó, Erika Tóth, Erika Szőcs, Mihály Cserepes, Katalin Nagy, Imre Kacskovics, Attila Patócs

**Affiliations:** 1Department of Molecular Genetics and the National Tumour Biology Laboratory, National Institute of Oncology, H-1122 Budapest, Hungary; 2Department of Oncology Biobank, National Institute of Oncology, H-1122 Budapest, Hungary; 3Hereditary Tumours Research Group, Hungarian Academy of Sciences, Semmelweis University, H-1089 Budapest, Hungary; 4Department of Laboratory Medicine, Semmelweis University, H-1089 Budapest, Hungary; 5Department of Anatomy, Semmelweis University, H-1094 Budapest, Hungary; 6Department of Pathology, National Institute of Oncology, H-1122 Budapest, Hungary; 7Department of Experimental Pharmacology, National Institute of Oncology, H-1122 Budapest, Hungary; 8ImmunoGenes-ABS Ltd., H-2092 Budakeszi, Hungary

**Keywords:** glucocorticoid receptor, glucocorticoid receptor alpha, glucocorticoid receptor beta, breast cancer, proliferation, migration, breast cancer progression, metastasis

## Abstract

**Background**. The dual role of GCs has been observed in breast cancer; however, due to many concomitant factors, GR action in cancer biology is still ambiguous. In this study, we aimed to unravel the context-dependent action of GR in breast cancer. **Methods**. GR expression was characterized in multiple cohorts: (1) 24,256 breast cancer specimens on the RNA level, 220 samples on the protein level and correlated with clinicopathological data; (2) oestrogen receptor (ER)-positive and -negative cell lines were used to test for the presence of ER and ligand, and the effect of the GRβ isoform following GRα and GRβ overexpression on GR action, by in vitro functional assays. **Results**. We found that GR expression was higher in ER− breast cancer cells compared to ER+ ones, and GR-transactivated genes were implicated mainly in cell migration. Immunohistochemistry showed mostly cytoplasmic but heterogenous staining irrespective of ER status. GRα increased cell proliferation, viability, and the migration of ER− cells. GRβ had a similar effect on breast cancer cell viability, proliferation, and migration. However, the GRβ isoform had the opposite effect depending on the presence of ER: an increased dead cell ratio was found in ER+ breast cancer cells compared to ER− ones. Interestingly, GRα and GRβ action did not depend on the presence of the ligand, suggesting the role of the “intrinsic”, ligand-independent action of GR in breast cancer. **Conclusions**. Staining differences using different GR antibodies may be the reason behind controversial findings in the literature regarding the expression of GR protein and clinicopathological data. Therefore, caution in the interpretation of immunohistochemistry should be applied. By dissecting the effects of GRα and GRβ, we found that the presence of the GR in the context of ER had a different effect on cancer cell behaviour, but independently of ligand availability. Additionally, GR-transactivated genes are mostly involved in cell migration, which raises GR’s importance in disease progression.

## 1. Introduction

In women, breast cancer is the most common cancer type worldwide (estimated 2.3 million new cases per year) [1]. Early-diagnosed breast cancer accounts for more than 90% of all cases, but despite the availability of modern treatment options, approximately one-third of these patients develop cancer recurrence/progression at a later time [2]. Locally advanced/metastatic breast cancer has a median overall survival of ~3 years, and the 5-year survival is only ~25% [3].

The optimal therapy is selected based on the immunophenotype of the tumour, determined by immunostaining of the oestrogen receptor (ER), progesterone receptor (PR), and human epidermal growth factor receptor 2 (HER2). Hormone receptors (ER and PR) are expressed in most (~75%) breast cancers, indicating the responsiveness to hormonal therapy, and their presence represents a better prognosis [4]. HER2 overexpression can be detected in ~15% of breast cancers due to gene amplification, and it is an important predictive marker for the response to anti-HER2 therapy. Additionally, HER2-enriched tumours are associated with a more aggressive clinical course and poorer prognosis.

Glucocorticoids (GCs), e.g., dexamethasone (dex), are routinely administered as adjuvant therapy to prevent hypersensitivity reactions and to manage the side effects of cytotoxic chemotherapy, due to their antiemetic and orexigenic effects. Besides their beneficial adjuvant impact, on the one hand, glucocorticoids were suggested to prevent breast cancer by decreasing the levels of various mediators, such as oestrogens, pro-inflammatory cytokines, and eicosanoids, potentially involved in the pathophysiology of breast cancer [2,4,5]. On the other hand, glucocorticoids might promote breast cancer progression by facilitating tumour cells to escape from immune surveillance, promoting metabolic dysfunction or insulin resistance [6,7,8,9,10,11]. An increased circulating GC level has been associated with breast cancer progression [7,12]. Additionally, in vivo animal models have also demonstrated that rats exposed to chronic stress (accompanied by increased GC levels in the blood) developed more aggressive breast cancer compared to non-stressed animals [13].

While in ER+ breast cancer the presence of GR has been reported to have a favourable prognosis, probably due to crosstalk between the two nuclear receptors [14], in ER− (and triple-negative) breast cancer, GCs supported cancer growth and metastasis leading to enhanced aggressiveness [14,15,16,17] (Figure 1). Additionally, in a translational study, glucocorticoids resulted in the activation of the glucocorticoid receptor during breast cancer progression and increased colonization, and reduced survival [7]. Additionally, the authors indicated that the judicious adjuvant administration of corticosteroids could be considered when treating cancer-related complications [7]. Due to the finding that GR can be activated in the absence of the ligand as well [18], the effect of the presence of the ligand on GR activity has an important relevance. Additionally, the potential beneficial role of GR antagonism has been suggested to increase apoptosis during chemotherapy efficacy in ER-negative breast cancers, blocking metastatic spread [9].

The association between systemic GC use and breast cancer risk was evaluated in a prospective cohort study by Cairat et al., including 62,512 postmenopausal women [19]. Overall, it was observed that the use of systemic GC exposure was not associated with overall breast cancer risk; however, it was associated with a higher risk of in situ breast cancer and a lower risk of invasive breast cancer. GC exposure was also inversely associated with the risk of stage 1 or stage 2 tumours, while it positively associated with the risk of stage 3/4 breast cancers [19]. In addition, Shi et al., described that GR negatively correlated with survival, and ER+ patients showed similar results compared to TNBC and invasive subtypes [20]. However, the literature data indicate that GR was not an independent predictor of survival, and no association was found between GR expression and breast cancer-specific survival (BCSS) or distant metastasis-free interval (DMFI) [14].

Additionally, we hypothesized that the presence of GR isoforms could be another explanation for the heterogeneous findings. The human GR is encoded by the *NR3C1* gene (nuclear receptor 3, group C, member 1). The gene itself is composed of nine exons, and different splice isoforms are generated by alternative splicing. GRα is considered to be the main and most abundant isoform in almost all tissues [21]. Besides GRα (“the classical receptor”), GRβ has been considered as the other main GR isoform differing in the splicing of exon 9. GRα and GRβ are identical up to amino acid 727. GRα consists of 777 amino acids, while in the GRβ protein, the 50 carboxy-terminal amino acids are replaced by 15 non-homologous amino acids, resulting in a protein of 742 amino acids [21]. As exon 9 encodes the ligand-binding domain, GRα and GRβ differ significantly in their ligand-binding abilities: GRβ is shorter, hence preventing GRβ from binding to the GCs. The GRβ isoform is also expressed ubiquitously among different tissues, but is detected at lower levels compared to GRα [22]. The relative expression levels of GRα and GRβ have been associated with GC sensitivity–insensitivity in various cell types. GRβ could induce GC resistance by forming a non-transactivating heterodimer with GRα, hence impairing GRα-mediated genomic actions, which is called the dominant-negative effect [22]. In addition to GRα-dependent mechanisms, GRβ has been also shown to have intrinsic activities, and it has been shown that it can regulate the activity of numerous genes related to the inflammatory process, cell communication, migration, and tumourigenesis in HeLa and U-2 OS, and in T24 bladder cancer cells [22] (Figure 1).

**Figure 1 cells-12-00784-f001:**
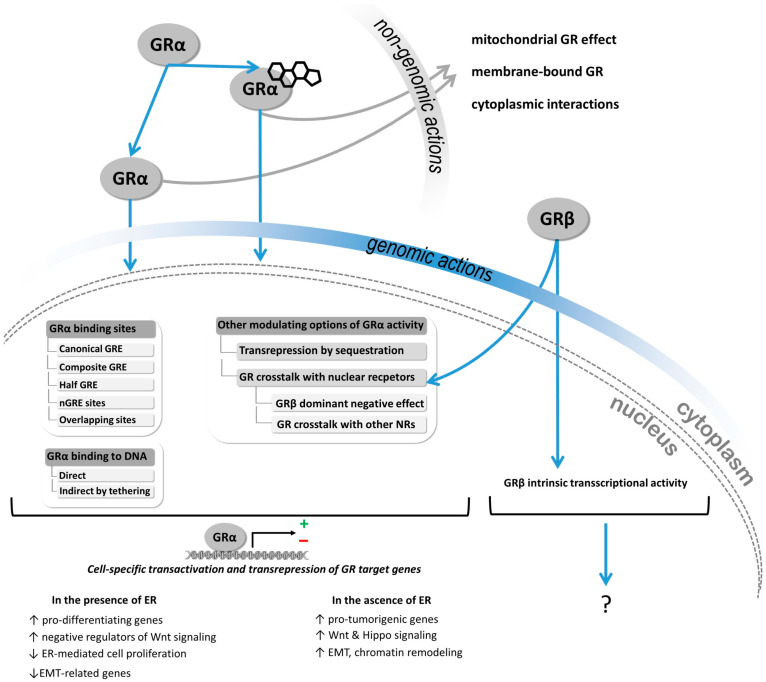
Mechanism of action of glucocorticoid receptor alpha (GRα) and beta (GRβ) in breast cancer cell. GR activity is strongly context-dependent, and determined by, among others, GR expression, splicing resulting in splice isoforms, posttranslational modifications and nuclear receptor crosstalk [8,9,10,23,24]. GRα activation can be ligand-dependent or -independent. By translocating into the nucleus, it binds to specific regulatory parts of the DNA (GR responsive elements, GREs) through which several genes’ expressions are induced or repressed in a cell type-specific manner. GRα and ER coactivation enhanced GRα binding to both GRE and oestrogen-responsive element (ERE), leading to an increased expression of pro-differentiating genes and negative regulators of pro-oncogenic Wnt signaling, and a decreased expression of epithelial–mesenchymal transition (EMT)-related genes. However, in the absence of ER, ligand-bound GRα binds to the GREs of several pro-tumourigenic genes, driving drug resistance and progression in TNBC (see details in the text, and in [6,23]). GRβ, due to its shorter sequence, cannot bind the ligand, but it is able to form a heterodimer with GRα. By binding to GREs, GRβ impairs GRα-mediated genomic actions, which is called the dominant–negative effect. While it has been described that GRβ is able to regulate proliferation and migration in other cell types, there is no clear evidence for its role in breast cancer development and progression (see details in the text, and in [22]).

While in the literature, several studies have reported significant associations between prognosis and GR expression [15,17,23,25], the detection of GR is still challenging. When investigating GR at the RNA level, in ER+ patients, high levels of GR expression in tumours have been found to be associated with a better prognosis compared to patients whose tumours harboured low levels of GR expression [15]. Additionally, high GR expression is associated with improved relapse-free survival in early-stage breast cancer patients [26]. In ER− patients, high levels of GR expression significantly correlated with shorter relapse-free survival independently of adjuvant chemotherapy [15]. Additionally, in ER− and triple-negative breast cancer patients, high GR expression was associated with a worse prognosis [16,17,25].

When GR is detected at the protein level, the findings are not so concordant. Shi et al. described that overall, GR negatively correlated with the survival rates in breast cancer patients, and ER+ patients showed similar results compared to TNBC and invasive subtypes [20]. Additionally, Adbuljabbar et al. observed that positive nuclear GR staining was associated with shorter breast cancer-specific survival in ER− and TNBC cases [14]. However, in this study, the authors indicated that GR was not an independent predictor of survival, and no association was found between GR expression and breast cancer-specific survival (BCSS) or distant metastasis-free interval (DMFI) in the whole series or the ER-positive group [14]. Additionally, Elkashif reported different outcomes in the context of anthracycline-based chemotherapy, depending on GR expression in ER− patients [25].

Based on all the ambiguous findings and the challenging detection of GR and GR isoforms, the roles of GCs and GR expression in breast cancer development and during progression are still diverse and context-dependent. Due to the unclear biological mechanism of action, we aimed to investigate the potential roles of (i.) different GR isoforms in breast cancer cell behaviour, since it is known that GRβ has an opposite effect compared to the most abundant isoform GRα [22,24], (ii.) the context of oestrogen receptor, and (iii.) the presence of receptor ligands as impacting the action of the glucocorticoid receptor.

## 2. Materials and Methods

### 2.1. Samples and Validation Datasets

We investigated GR protein expression in 20 independent patients with breast cancer (9 triple-negative (TNBC) and 11 luminal A type ER+) through the Department of Pathology at the National Institute of Oncology, Hungary. Pathological assessment (histology and immunostaining for oestrogen, progesterone, Her2 receptor status, and Ki67 proliferation indices) were done as part of the routine diagnostics that gave the basis of breast cancer subtype classification according to [4]. Control samples were selected from an FFPE block of a surgical specimen of an ER+ breast cancer patient where no malignant tissue was identified by the pathologist in parts adjacent to the tumours. The study was approved by the Scientific and Research Committee of the Medical Research Council of the Ministry of Health, Hungary (BMEÜ/1774-1/2022/EKU). Histologic characteristics of samples used for GRtotal and GRβ immunohistochemistry can be found in Appendix A.

Different validation sets of GR protein and the encoding *NR3C1* gene expression were investigated in normal breast tissue, breast cancer, and other cancer types through the Protein Atlas database (https://www.proteinatlas.org/ (accessed on 26 October 2022)). GR protein expression was evaluated in 184 normal tissue samples (see details of Figure 2) by immunohistochemistry using GRtotal antibodies (Cat#HPA004248, Atlas Antibodies, RRID:AB_1078976; and Cat#sc-8992, Santa Cruz Biotechnology, RRID:AB_2155784). *NR3C1* gene expression in normal breast specimens was also tested in 459 (168 females and 291 males) samples.

In different types of cancer, *NR3C1* expression was investigated in 7931 specimens (see details in Figure 3). The GR protein in breast cancer was examined in 16 breast cancer samples using the same antibodies as in normal tissues.

Gene expression and mutational data of *NR3C1*, *BRCA1*, *BRCA2*, *PTEN*, and *TP53* in 86 breast cancer cell lines of the Cancer Cell Line Encyclopedia (CCLE) were used to assess genetic dependency through the DepMap portal (https://depmap.org/portal/ (accessed on 26 October 2022))

For exhaustive gene correlation analysis, the bc-GenExMiner v4.8 database of published annotated breast cancer transcriptomic (DNA microarrays (*n* = 11,359) and RNA-seq (*n* = 4421)) data were used (PMID: 23325629) (accessed on 11 November 2022). Gene set enrichment analysis was applied for the functional annotation of selected gene sets. Common terms for gene ontology biological processes were further analyzed by MonaGO [27], redundancy was reduced and similar GO terms were clustered. Table 1 summarizes different sample cohorts used in this study.

### 2.2. Cell Cultures, Treatments, and Transfections

Two human triple-negative (MDA-MB231 (#92020424) and HS578T (#86082104)) and two oestrogen-positive breast cancer cell lines (T47D (#85102201) and ZR-75-1 (#87012601)) were purchased from European Collection of Cell Cultures (ECACC) General Cell Collection. Cell lines were propagated at 37 °C in a humidified atmosphere containing 5% CO_2_ and used until passage number 25.

HS578T cells were grown in Dulbecco’s modified eagle’s medium (DMEM; #10-013-CV, Corning, Corning, NY, USA) supplemented with 10% fetal bovine serum (FBS; #P40-37500 PAN-Biotech, Aidenbach, Germany) and 1% penicillin/streptomycin (#10378016, Thermo Fisher Scientific, Waltham, MA, USA). MDA-MB-231, T47D, and ZR-75-1 breast mammary gland carcinoma cell lines were maintained in RPMI-1640 base medium (#BE12-702F, Lonza Biosciences; Basel, Switzerland) using fetal bovine serum (FBS; #P40-37500 PAN-Biotech, Aidenbach, Germany) in a final concentration of 10% and 1% penicillin/streptomycin (#10378016, Thermo Fisher Scientific, Waltham, MA, USA).

Three times a week, the cell culture medium was replaced with a fresh complete medium. When cells reached 90% confluence, they were detached from the bottom of the flask using 0.05% Trypsin-EDTA (#25300062, Invitrogen, Thermo Fisher Scientific, Waltham, MA, USA). Microscopic control and imaging were done with an EVOS M7000 imaging system using ×10 objective.

In experimental settings, cells were kept in complete media or steroid-free media for 48 h before plating. Steroid-free media was prepared using charcoal-stripped FBS as previously reported [28]. Then, the cells were plated on 6-well tissue culture plates (maintaining complete or steroid-free conditions) and after 24 h they were transfected as described below. All experiments were carried out three times.

Cells were seeded in 6-well plates in antibiotic-free media 24 h before transfection. For transfections, pcDNA3.1(+) expression vectors containing cDNA of GR-α (pcDNA3.1-GR-α) and GR-β (pcDNA3.1-GR-β) were used as previously reported [29]. An empty vector (pcDNA3.1) was utilized as the control plasmid per transfection in the Western blot (WB). For transfections, Lipofectamine 3000 (#L3000001, Invitrogen, Thermo Fisher Scientific, Waltham, MA, USA) was used following the manufacturer’s instructions. Cells were harvested or fixed 24 to 48 h post-transfection.

### 2.3. Western Blot

After transfection, the cells were lysed in M-PER™ Mammalian Protein Extraction Reagent (#7851, Thermo Fisher Scientific, Waltham, MA, USA) supplemented with halt protease inhibitor cocktail (#87785, Thermo Fisher Scientific, Waltham, MA, USA). The total protein concentration was determined with the Bradford Protein Assay (#6916, Merck, Darmstadt, Germany) using bovine serum albumin as standard (#A9418, Merck, Darmstadt, Germany). In total, 20 µg samples of protein homogenates were loaded onto the polyacrylamide gel. Then, the proteins were transferred onto a polyvinylidene difluoride (PVDF) membrane (#88518, Thermo Fisher Scientific, Waltham, MA, USA). Membranes were blocked in blocking buffer (tris-buffered saline with Tween 20 solution (TBST) containing 5% non-fat dry milk) for 1 h at room temperature. The blots were then incubated with primary antibodies Rabbit Polyclonal Glucocorticoid Receptor antibody (#GTX101120, GeneTex, Irvine, CA, USA), 10G8 (ImmunoGenes Ltd., Budakeszi, Hungary), and beta-actin (#4967, Cell Signaling Technology, Danvers, MA, USA) in 1:1000 dilution overnight at 4 °C. The following day, the blots were incubated with goat anti-mouse immunoglobulins/HRP (#P0447, Dako, Santa Clara, CA, USA) and goat anti-rabbit immunoglobulins/HRP (#P0448, Dako, Santa Clara, CA, USA) secondary antibodies in 1:1000–1:2000 dilution for 1 h at room temperature. The proteins were visualized using the SuperSignal West Pico PLUS chemiluminescence detection kit (#34577, Thermo Fisher Scientific, Waltham, MA, USA).

### 2.4. Immunohistochemistry

For immunohistochemistry analysis, paraffin-embedded sections were processed by the ABC technique to visualize antigens (ABC Elite Kits, Vector Laboratories, Burlingame, CA, USA), slightly modifying a previously described protocol [30]. For single immunohistochemistry, ABC immunoperoxidase staining and a DAB solution as the chromogen (Vector Laboratories, Burlingame, CA, USA, Impact^®^ DAB Substrate, Peroxidase (HRP) were applied. To visualize the glucocorticoid β receptor a mouse monoclonal antibody produced and characterized by ImmunoGenes Ltd. (10G8) was used at a dilution of 1:4000. For localizing the total glucocorticoid receptor (α and β), a polyclonal antibody against human GR N terminal (GTX101120, GeneTex, Irvine, CS, USA) was applied at a dilution of 1:100.

Briefly, the paraffin sections were dewaxed and rehydrated with 0.05 mol/L potassium phosphate-buffered saline (KPBS). The primary antibodies were applied for 1 h at room temperature, followed by 24 h at 4 °C (diluted in KPBS + 0.4% triton-X100). On the second day, the samples were incubated with biotinylated goat anti-mouse antibody (BA-9200 Vector Laboratories, Burlingame, CA, USA) or anti-rabbit (BA-1000, Vector Laboratories, Burlingame, CA, USA) for 1 h in room temperature at a 1:500 dilution in KPBS + 0.4% triton-X100. Then, sections were incubated with ABC solutions for 1 h at room temperature (45 mL each A and B in 10 mL of KPBS + 0.4% triton-X100, Vector Laboratories, Burlingame, CA, USA). The samples were then rinsed three times for 5 min each in KPBS and then exposed to DAB H2O2-containing chromogen solution. Staining was performed for 8 min in the case of the GR β antibody and 12 min for the GRtotal and was terminated by rinsing in KPBS. Sections were counterstained with hematoxylin (Novolink, Leica Biosystems Newcastle Ltd., Newcastle Upon Tyne, UK) and coverslipped with Glycergel aqueous mounting medium (Agilent, Santa Clara, CA, USA).

### 2.5. Immunocytochemistry

Cells seeded on coverslips were washed twice with PBS and then fixed with 4% paraformaldehyde (PFA) in PBS. Coverslips were blocked in 5% bovine serum albumin in PBST at room temperature for 1 h, then incubated with primary GRβ antibody (ImmunoGenes Ltd., #10G8) at 1:100 dilution overnight at 4 °C. Goat anti-mouse IgG (H+L) highly cross-adsorbed the secondary antibody, and Alexa Fluor Plus 555 (Thermo Scientific #A32727) was applied as the secondary antibody (at 1:500 dilution) for 1 h at room temperature. The cells then were incubated with Hoechst 33342 to stain the nuclei. Images were obtained using a 10× objective.

### 2.6. Cell Viability, Proliferation, Live–Dead Cell Ratio, and Cell Migration

Cells were seeded on 6-well plates. Cell viability, proliferation, and dead cell ratio were investigated as we previously reported [31]. Briefly, for cell viability assessment, the metabolic Alamar Blue assay (#DAL1025, Invitrogen, Thermo Fisher Scientific, Grand Island, NY, USA) was used. Fluorescent signals were detected using a flash spectral scanning multimode reader (#5250040, Varioskan, Thermo Fisher Scientific, Waltham, MA, USA) with SkanIt Software 2.4.5 RE (ex: 560 nm, em: 590 nm). This metabolic assay is applied as a cell health indicator using the metabolic activity of living cells to quantitatively measure viability. Optical density (OD) data were presented as normalized values relative to monolayer cultures at each point in average ratio ± standard deviations. To assess cell proliferation cell numbers were determined using 0.4% Trypan Blue staining (#15250061, Gibco, Thermo Fisher Scientific, Waltham, MA, USA). Trypan Blue staining represents a cruder analysis to identify dead cells. Results from Trypan Blue assays (live cell number) have been defined as “proliferation”. All experiments were repeated at least three times (biological replicates) with one to three technical replicates in each experiment. Mean and standard deviation were calculated and are illustrated on graph bars.

To assess the effects of GRα and GRβ on migration, wound-healing assays were performed on 24-well plates as previously reported [32]. Twenty-four hours after transient transfection the cell monolayer was wounded using a 200 μL pipette tip and floating cells were washed with phosphate-buffered saline (PBS) (#21-040-CV, Corning, Corning, NY, USA). Photos were taken after 0, 24, and 48 or 0, 6, and 12 h depending on cell type. Images were analyzed with ImageJ Software (https://imagej.nih.gov/ij/ (accessed on 7 January 2022), Bethesda, MD, USA) to calculate cell-free area (CFA %: [(CFA at target time/CFA 0 h) × 100]) [32].

### 2.7. Statistical Methods

For the comparison of multiple groups, analysis of variance was used to identify statistical significance among different groups, and the Dunnett test was used to correct for multiple comparisons. To compare the two groups unpaired t-test with Welch’s correction was applied. A *p*-value < 0.05 was considered statistically significant. For investigating the correlation between NR3C1 and other genes’ expression in RNAseq studies, Pearson’s correlation was used.

## 3. Results

### 3.1. Characterization of Glucocorticoid Receptor Expression in Normal and Cancerous Breast Tissue and Breast Cancer Cell Lines

As a first step, we examined the glucocorticoid receptor encoding *NR3C1* expression at the RNA and GR protein levels across different normal tissues and cancer types, including normal and cancerous breast samples, using high-throughput data (Figure 2A,B). Expectedly, due to its general function, *NR3C1* showed an overall broad expression among different tissue types, hence low tissue specificity. In normal breast tissue, glucocorticoid receptor protein expression was found to be medium and high compared to other tissue types (Figure 2A). Interestingly *NR3C1* expression was higher in male breast tissue compared to female; however, it did not depend on age in either group (Figure 2B).

Regarding breast cancer, *NR3C1* expression was found around the average level compared to different tumour types at the RNA level (Figure 3A). Regarding glucocorticoid receptor protein immunohistochemistry in human breast cancer tissues, two different types of anti-GR antibodies (HPA004248, CAB010435) showed variant staining—some only nuclear, some cytoplasmic/membranous—and nuclear staining on the same samples of the Protein Atlas database (Figure 3B). In contrast to normal breast tissue, *NR3C1* expression was decreased in male breast cancer tissue compared to females, reaching the level of significance (*p* = 0.055); however, it did not depend on age in either group (Figure 3C).

As a further step, we also analyzed *NR3C1* expression in 86 different breast cancer cell lines. *NR3C1* did not show a significant expressional difference between primary and metastatic breast cancer cell lineages; however, its level was higher in ER− samples compared to ER+ cases (Figure 3D). *NR3C1* expression was independent of *BRCA1*, *BRCA2*, *PTEN*, or *TP53* mutational status (data not shown).

### 3.2. Different Glucocorticoid Receptor Isoforms in Breast Cancer

GRα and β isoforms have important roles with opposite functions, hence we assessed the expression of the GR using an N-terminal specific antibody referred to as GRtotal, and a selective antibody specific against the GRβ isoform (GRβ) on an independent sample cohort of 9 TNBC and 11 luminal A type (ER+, PR+, negative for HER2) breast cancer tissues (see primary staining and negative controls omitting the primary antibody in Appendix A, which also indicates the specificity of cytoplasmic localization). We found that both GRtotal and GRβ were detectable in normal and cancerous breast specimens (Figure 4). Staining was heterogeneous among samples irrespective of tumour type. Both GRα and GRβ exhibited mostly cytoplasmic localization in tumours, and in some samples with nuclear positivity (Figure 5).

When dissecting different cell types, in control tissue, the mammary gland lactiferous duct epithelial cells showed less frequent immunostaining with GRβ than with GRtotal. Both isoforms could appear as nuclear and cytoplasmic as well. In almost every epithelial and myoepithelial cell, the nuclear staining of GRtotal could be observed. GRβ labeled a few of the epithelial and myoepithelial cells. Other connective tissue cells, such as fibrocytes, adipocytes, and endothelial cells, also showed GRtotal positivity and much less GRβ positivity. The oestrogen-positive and -negative cancer tissues showed both cytoplasmic, and less frequently, nuclear staining for GRtotal and GRβ. Generally, the infiltrating lymphocytes exhibited intensive GRtotal and GRβ staining, but not in all lymphocytes.

The specificity of the GRβ antibody was tested by in vitro models using expression vectors encoding GRα and GRβ isoforms without any cross-reactivity of GRβ with GRtotal (Figure 6A). In line with our immunohistochemical findings, we found that GRβ localized mainly in the cytoplasm using immunocytochemistry in both control and transfected cells (Appendix A). We found significantly higher GRtotal expression in TNBC cell lines compared to ER+ ones, which is in line with the finding at the RNA level. Additionally, compared to GRα, a low amount of GRβ was detected in both ER+ and ER− cell lines (Figure 6B,C)

### 3.3. The Opposite Effect of Glucocorticoid Receptor Expression in Breast Cancer Cell Viability, Proliferation, Cell Death, and Migration Depending on Hormone Receptor Status

It has been previously reported that GR expression represented a worse prognostic factor for ER−, but not for ER+ patients, and that GRβ has an opposite effect compared to the main GR isoform (GRα). Therefore, we separately investigated the effects of GRα and GRβ on breast cancer cell behavior. In parallel, we assessed the effect of the presence of the receptor ligands on glucocorticoid action, as well as in the context of the oestrogen receptor.

We found that GRα increased cell viability and cell proliferation in ER− cells independently of the presence of the ligand, while it had no or a mild effect on ER+ breast cancer cells regardless of the availability of steroid ligands (Figure 7A,B). GRβ expression did not alter cell viability or proliferation in either type of cell. Interestingly, GRβ increased the dead cell ratio in ER+ but not in ER− cells, and this effect was also independent of the presence of the ligand (Figure 7C). Both GRα and GRβ increased the cell migration of ER− breast cancer cell lines, and neither of them influenced the cell migration of ER+ cells (Figure 8A–C and Figure 9A–C).

Based on the finding that GR signalling did not depend on the presence of the ligand, we screened genes that exhibited a significant positive and negative correlation with *NR3C1* in breast cancer tissue samples (10,455 samples analyzed by 57 microarray studies and 4421 samples analyzed by three RNAseq studies). We assessed biological functions via GO biological process gene set enrichment analyses of both microarray and RNAseq experiments and focused on the common findings. We found that genes that positively correlated with NR3C1 were mainly implicated in the cell migration, angiogenesis, and intracellular steroid hormone receptor signaling pathways (Figure 10 and Table 2).

Negatively correlated genes represented smaller gene sets compared to positively correlated genes: 15% and 3% of all correlating genes in microarray and RNAseq studies, respectively. Therefore, we investigated the union of the biological functions of the negatively correlated genes that were involved in cell division and ubiquitination (Table 3). These findings corroborate our in vitro results as well.

## 4. Discussion

The dual (tumour-suppressing and -promoting) role of GCs has been well-documented [6]. In animal models, GCs protected against cancer development, and studies have indicated the tumour-suppressive roles of GR in epithelial solid cancers [6,33]. However, GR action in cancer biology appears to be strongly cell type- and context-dependent [12,23].

Due to its essential function in homeostasis, GR is abundantly expressed among different tissue and cancer types. In normal breast tissue, the reasons for and relevance of our finding that GR showed higher expression in males compared to females need further investigation. In contrast to normal tissue, in breast cancer, *NR3C1* showed the opposite—its expression was decreased in male compared to female patients. The lower expression of *NR3C1* in male breast cancer seems to be in line with the finding that breast cancer in males is mostly oestrogen-positive, and it has a good prognosis [3,34].

### 4.1. Challenging Detection of GR Expression in Breast Cancer

In line with the findings of others, we detected GR protein expression in the majority of breast tissues. We also observed that *NR3C1* gene expression was increased in ER− breast cancer cell lines compared to ER+ ones. On the protein level, both GRtotal and GRβ were detectable in normal and cancerous breast specimens. Staining was heterogeneous among samples irrespective of tumour subtype (i.e., presence of oestrogen receptor). Both GRα and GRβ exhibited mostly cytoplasmic localization in tumour samples, and in some samples with nuclear positivity. Our results regarding GRtotal are similar to the findings described in the Protein Atlas using the CAB010435 antibody (Cat#sc-8992, Santa Cruz Biotechnology) that also indicated both cytoplasmic and nuclear staining. However, when another antibody (HPA004248: Cat#HPA004248, Atlas Antibodies) was used, only nuclear staining was indicated. While nuclear positivity indicates GR activation and cytoplasmic positivity reflects the expression and non-genomic action of GR, the reliable detection of GR is considered crucial when correlating with clinicopathological parameters.

The ambiguous results could be due to both technical and biological factors. In immunohistochemistry staining, various antibodies were used, and different staining patterns (purely nuclear vs. cytoplasmic/nuclear) have been observed, which could be traced back to the possibility of technical (e.g., antigen retrieval and antibody specificity) aspects, hence warranting caution in the interpretation of results.

### 4.2. GR Expression in Breast Cancer in the Context of ER

GR and ER coactivation enhanced GR binding to both glucocorticoid-responsive elements (GRE) and oestrogen-responsive elements (ERE), resulting in anti-tumourigenic effects, such as the increased expression of pro-differentiating genes and negative regulators of pro-oncogenic pathways, as well as the decreased expression of EMT-related genes [35]. As GR and ER co-occupy the same genomic nuclear receptor-responsive regions, GCs antagonized oestrogen-stimulated endogenous ER target gene expression and oestrogen-mediated cell proliferation [23,35,36,37]. On the other hand, oestrogen also influences GC action. Oestrogen could induce the dephosphorylation of GR, consequently decreasing its activity on the target genes involved in cell growth arrest [38]. Additionally, ER antagonists could lead to the enhanced proteasomal degradation of GR [39]. This GR–ER crosstalk manifested as the improved relapse-free survival of breast cancer patients with ER-positive tumours, and GR was related to a favourable prognosis, while low GR expression was associated with worse outcomes, such as high Ki67, p53, and CD71 expression [35,40].

According to the context of ER, we showed that GR expression was higher in ER− breast cancer cells compared to ER+ ones, which may indicate a potential reciprocal inhibitory action between GR and ER [41]. Additionally, our findings that the presence of GR itself increased cell proliferation in ER− breast cancer cells, while it had no impact on ER+ tumour cells, are in line with studies reporting an association between GR expression and prognosis/outcome [15,16,17,25].

The GR–ER crosstalk is additionally illustrated by feedback loops, where GR could back-regulate ER expression. While there is no GRE identified in the promoter of ER, the indirect regulation of ER by GR can be hypothesized as a feedback loop control. The promoters of ERα contain multiple predicted and validated transcription factor-binding sites [42,43]. Several of them are in indirect interaction with GR, such as ER itself, *BRCA1*, *ZEB2*, NF-κB, and circadian genes [42,43]. In addition, *DNMT1* and *ZEB1*, as GR-regulated genes, can induce ERα promoter methylation and the down-regulation of ERα expression [42]. Similarly, histone acetylation and methylation also play a role in ER expression, while histone acetyltransferases and demethylase are also regulated by GR at the promoter level [43]. Additionally, another way in which GR and ER signaling interact is by decreasing levels of free oestrogen through the GR-mediated activation of oestrogen sulfotransferase [44].

In the complex interaction network of GR–ER, there are indirect processes in which ER is itself regulated by other receptors, which, in turn, could regulate GR expression. Signal transduction by Her2 and epidermal growth factor receptors (EGFR) was described to alter the phosphorylation of ER and ER-dependent signaling irrespective of the presence of ER ligands [45]. In addition, both oestrogen and growth factor signaling pathways regulate the secretion of vascular endothelial growth factors that stimulate tumour-associated angiogenesis [45]. Additionally, evidence has suggested that crosstalk between ER and growth factor receptor pathways contributed to the development of tamoxifen resistance in breast cancer. Signaling via the EGFR and Her2 could activate both ER and the ER coactivator AIB1. In turn, ER located in the cell membrane can activate the growth factor receptor pathways [46].

Besides interactions between nuclear receptors, crosstalk between GR and growth factors has also been reported [47,48]. EGFR, one of the most active growth factors exerting strong growth-promoting effects in the mammary epithelium [47], can interact with GR through both genomic and epigenomic processes [48]. Regarding crosstalk with other growth factors, GR has been shown to be a required effector of TGFβ1-induced p38 MAPK signaling [49], and it suppresses the transcription of the insulin receptor substrate 1 (IRS-1), which mediates insulin-like growth factor (IGF) signals [20].

### 4.3. The Role of GRβ Isoform in GR Action in Breast Cancer

Interestingly, we found a similar effect of GRα and GRβ following transfection regarding viability and proliferation. Our data suggest that in ER– breast cancer cells, even the increased relative expression of GRβ does not abolish the effect of GRα regarding tumour cell viability, proliferation and migration. This finding was somewhat surprising, as in allergic respiratory and inflammatory bowel diseases, increased GRβ has been associated with resistance against glucocorticoids [22].

However, GRβ increased the dead cell ratio in ER+ cells only, while it had no—or a mildly opposite—effect in cells lacking ER. While the crosstalk between GRα and ER is well-known [35], the explanation of the different effects of GRβ depending on the presence of ER needs further clarification.

### 4.4. The Effect of Ligand Availability on GR Action on Breast Cancer Cell Behavior

In the physiological GR action, in the presence of steroid ligand, the GR monomers are removed from their GRE half sites, and instead, GR-dimer formation and assembly on classical GREs in the DNA occurs [23]. Indeed, in breast cancer, unliganded GR has been described to play a protective role. In non-malignant mammary cells, GR has been shown to bind to the promoter region of the *BRCA1* gene, up-regulating its expression [50]. GCs induced a loss of GR recruitment to the *BRCA1* promoter with a concomitant decrease in *BRCA1* expression [50,51]. Despite this interaction of GR with *BRCA1* expression, we did not find any effect of *BRCA1* (or other hereditary breast cancer predisposition genes) mutation status on GR expression.

Interestingly, the effects of both GRα and GRβ on cell viability, proliferation, dead cell ratio, and cell migration were independent of the presence of the ligand, indicating that the receptor expression/the presence of the receptor itself may have an important prognostic role.

The genomic effects of GRα include both transactivation and transrepression, which could be realized by the direct binding of GRα to GRE sequences. Several pieces of evidence have substantiated that GRα can also be activated in the absence of ligands [6,50,51,52,53]. Indeed, certain chemicals, elevated temperature, cellular pH, and shear stress were demonstrated to induce GRα nuclear translocation, hence its activation [21,52]. Additionally, posttranslational modifications of the receptor and the presence of TNFα were also shown to induce ligand-independent GRα activation [21,54]. Moreover, non-genomic GC action (e.g., cytoplasmic, membrane-bound, or mitochondrial GR action) could also occur independently of the ligand [23].

Furthermore, the GRβ negative–dominant effect on GRα can occur in a ligand-independent way. Upon GRE binding, GRβ competes with GRα, or it forms an inactive heterodimer; consequently, it does not induce transcription [6,47,52,55]. It is also suggested that GRβ can bind other ligands (e.g., synthetic GC antagonists, unknown molecules, or endogenous steroids) as well. Moreover, the intrinsic activity of the GRβ isoform (also in the absence of the ligand) has been proven by in vitro and in vivo experiments, where GRβ exerted transcriptional activity on several genes, including both GRE-containing promoters and non-GC-regulated genes [6,47,52,55].

Based on our findings, ligand-independent GR action (including both GRα and GRβ) may play an important role in breast cancer cell proliferation and migration.

### 4.5. GR Activity Signature in Breast Cancer

GR transactivates or transrepresses (in an either ligand-dependent or -independent way) numerous genes. Additionally, the GR activity signature (expressional changes) was demonstrated to have a stronger association with RFS than GR expression alone [56]. Therefore, we screened for genes positively and negatively correlating with GR in breast cancer specimens from 14,876 patients. We found that positively correlated (transactivated) genes were implicated mainly in cell migration, and also in the angiogenesis and intracellular steroid hormone receptor signaling pathways, while they were negatively correlated (transrepressed) with cell division and ubiquitination. These biological processes of GR action are fully reflected by our results, which were derived by in vitro functional assays.

Previously, GR activation has been linked to apoptosis regulation and the modulation of the expression of apoptotic genes by interfering with p53 function in ER+ breast cancer [36,57,58]. Furthermore, GR activation was protective against apoptosis both in vitro and in vivo [59,60], with which our data—demonstrating GRβ’s effect on the dead cell ratio in ER+ breast cancer cell lines—are in complete agreement.

Additionally, in TNBC, the GR activation signature was also related to epithelial–mesenchymal transition (EMT), cell adhesion, and inflammation pathways [56,61]. Recently, Obradovic et al., while investigating both patient-derived and TNBC cell line-derived xenograft models, demonstrated that GR activation increased breast cancer heterogeneity and metastasis. In this study, elevated GC levels during cancer progression augmented tumour cell colonization and reduced the survival of animal models of ER-negative breast cancer [7], reflecting our results on the cell migratory GR signature.

## 5. Conclusions

GR’s role in cancer biology is still ambiguous. This is most probably a consequence of the strong context-dependent activity of GR. Indeed, the role of altered GR expression, different isoforms due to alternative splicing, posttranslational modifications, availability of the ligand and nuclear receptor crosstalk have been suggested to modulate GR action in steroid-sensitive tissues and diseases (e.g., asthma, inflammatory bowel diseases), and in cancer as well.

Therefore, in this study, we aimed to unravel the context-dependent function of GR action in breast cancer, such as the presence of ER, the role of GRβ isoforms, and the availability of the ligand.

We have reinforced the finding that the expression of ER is a main factor in GR action probably due to receptor crosstalk. We found that the main isoform, GRα, increased cell proliferation and viability in ER− (TNBC) cells. By dissecting the effects of GRα and GRβ, we have demonstrated that GRβ showed low/heterogenous abundance in breast cancer, and that it has a similar effect on breast cancer cell lines’ viability, proliferation, and migration. However, the GRβ isoform has an opposite effect depending on the presence of ER, increasing dead cell ratio in ER+ breast cancer cells compared to ER− ones. Interestingly, we found that GRα and GRβ’s effects on cell viability, proliferation, dead cell ratio, and cell migration did not depend on the presence of the ligand, suggesting the role of the “intrinsic”, ligand-independent action of GR in breast cancer.

Our findings may add a new perspective regarding the previously suggested potential danger of adjuvant steroid therapy. Furthermore, different GR isoforms may have an important effect on the outcome of ER+ breast cancer patients by increasing dead cell ratio. These data add a further degree of complexity to the context-dependent effects of glucocorticoid and GR action in breast cancer.

## Figures and Tables

**Figure 2 cells-12-00784-f002:**
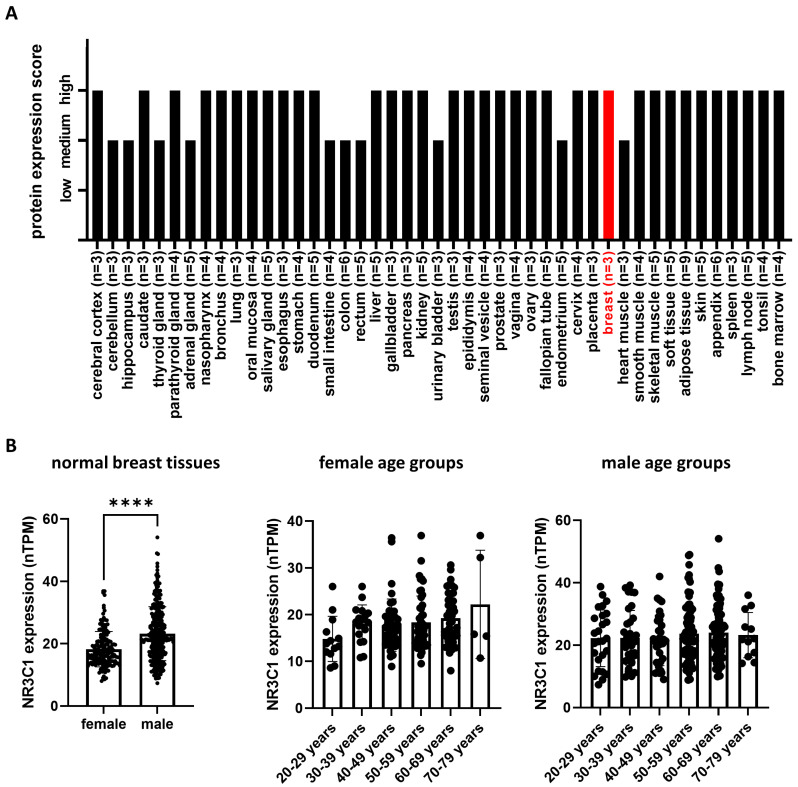
Glucocorticoid receptor expression characterization in different normal tissue types (**A**) and normal breast tissue (**B**). ****: *p* < 0.0001.

**Figure 3 cells-12-00784-f003:**
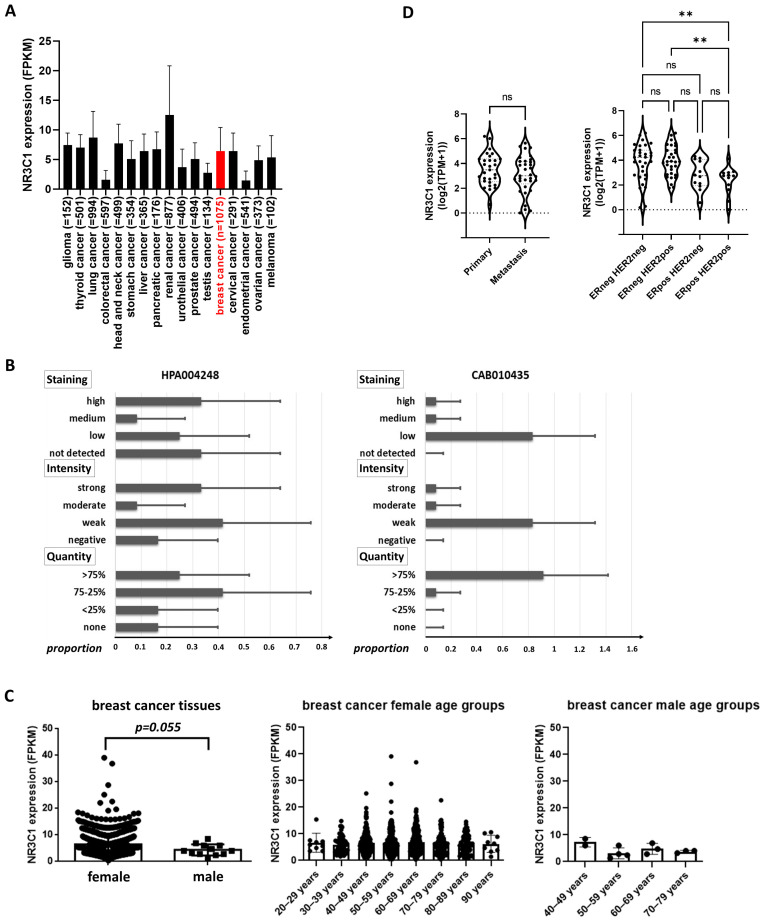
Glucocorticoid receptor expression in different cancer tissue types (**A**). (**B**) Immunostaining characteristics of GR using two commercially available antibodies (HPA004248: Cat#HPA004248, Atlas Antibodies; CAB010435: Cat#sc-8992, Santa Cruz Biotechnology). (**C**) Glucocorticoid receptor expression in female and male breast cancers; (**D**) GR encoding *NR3C1* gene expression in primary vs. metastatic and in different subtypes of breast cancer cells. **: *p* < 0.01; ns: not significant.

**Figure 4 cells-12-00784-f004:**
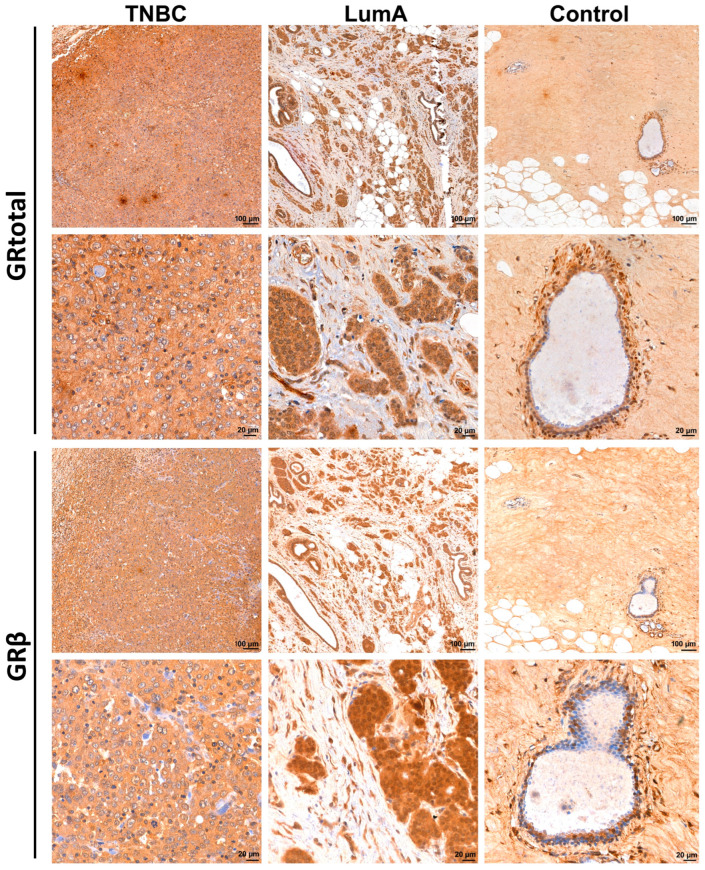
Representative images of glucocorticoid receptor protein staining in normal breast (control), triple-negative (TNBC) and oestrogen-positive (luminal A type, LumA) breast cancer, counterstained by hematoxylin. Tumour tissues show great variance in terms of the immunostaining pattern of GR. We show here the most intensively stained samples from both TNBC and LumA in each group.

**Figure 5 cells-12-00784-f005:**
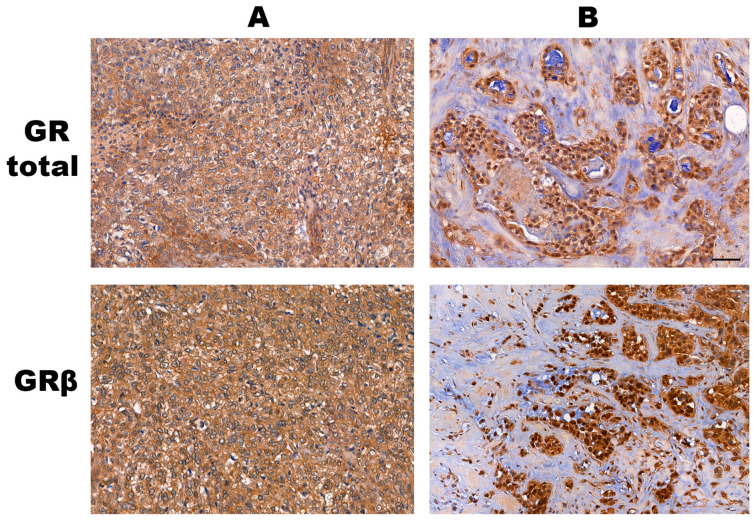
Examples of the cytoplasmic (panel **A**) and mixed cytoplasmic+nuclear (panel **B**) staining patterns of the GRtotal and GRβ proteins. In Panel **B**, some nuclei are positive and some of them are negative for staining in the tumour tissue. The line indicates 50 μm. On this representative image, all tumours are of the triple-negative subtype.

**Figure 6 cells-12-00784-f006:**
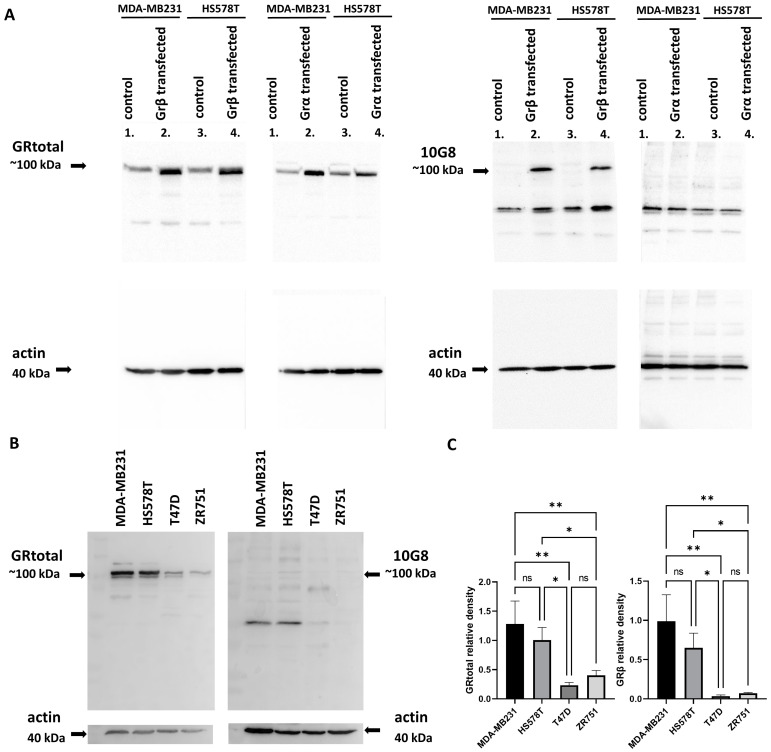
(**A**) Discrimination of GRtotal and GRβ isoforms using Western blot. (**B**) Representative images of GRtotal and GRβ endogenous expression in triple-negative and ER+ breast cancer cells. (**C**) Densitometry of GRtotal and GRβ Western blot performed on triple-negative (S578T and MDA-MB231) and ER+ breast cancer cells (T47D and ZR751). Relative densities indicate GRtotal/actin and GRβ/actin values. *: *p* < 0.05; **: *p* < 0.01; ns: not significant.

**Figure 7 cells-12-00784-f007:**
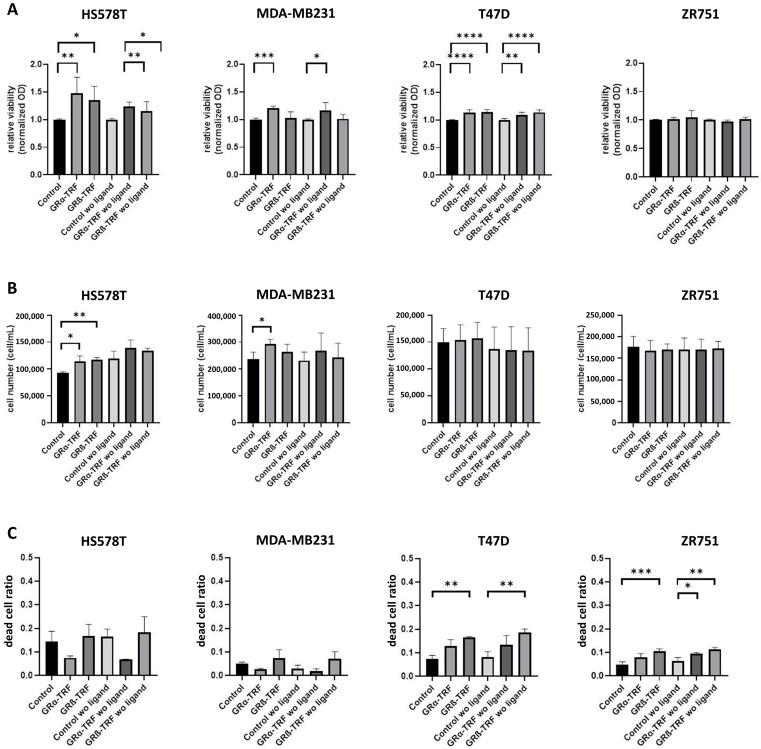
Investigating viability (**A**), cell proliferation (**B**), and dead cell ratio (**C**) following GRα and GRβ transfection in the presence or absence of the ligand in triple-negative (HS578T and MDA-MB231) and ER+ breast cancer cells (T47D and ZR751). *: *p* < 0.05; **: *p* < 0.01; ***: *p* < 0.001; ****: *p* < 0.0001.

**Figure 8 cells-12-00784-f008:**
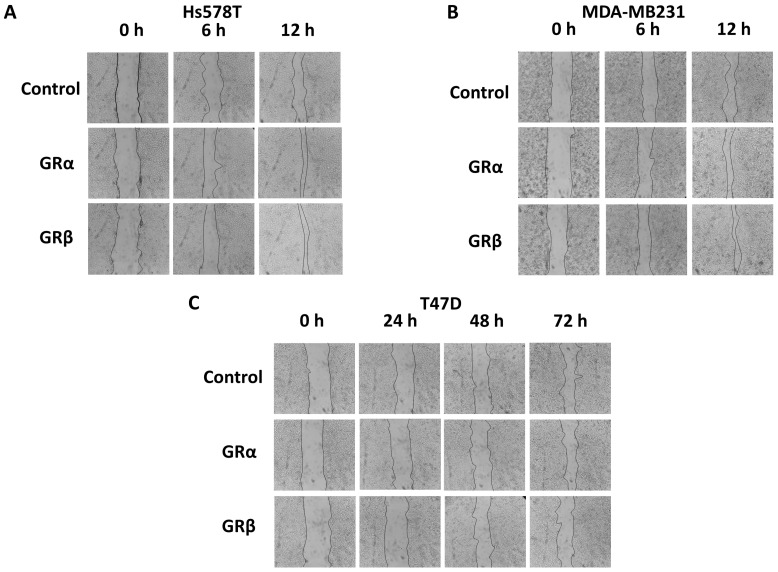
Representative cell migration images of triple-negative ((**A**) HS578T and (**B**) MDA-MB231) and ER+ breast cancer cells ((**C**) T47D) following GRα and GRβ overexpression. Photos were taken using 3.2× objective at 6–12 and 24–72 h following wounding.

**Figure 9 cells-12-00784-f009:**
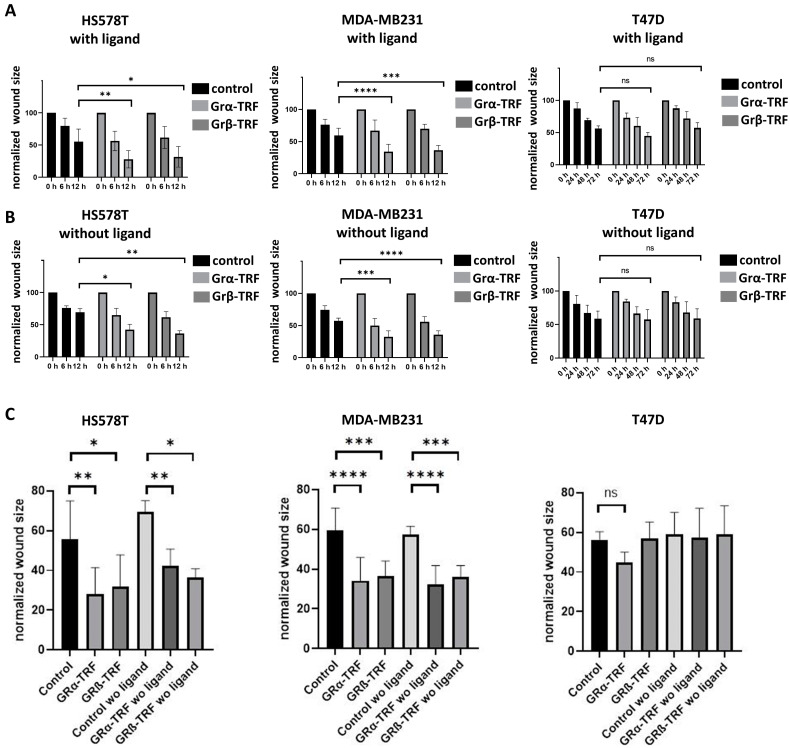
Time-lapse (**A**,**B**) and comparative (**C**) results of cell migration of triple-negative ((**A**) HS578T and (**B**) MDA-MB231) and ER+ breast cancer cells ((**C**) T47D) following GRα and GRβ overexpression in the presence and the absence of the ligand. *: *p* < 0.05; **: *p* < 0.01; ***: *p* < 0.001; ****: *p* < 0.0001; ns: not significant.

**Figure 10 cells-12-00784-f010:**
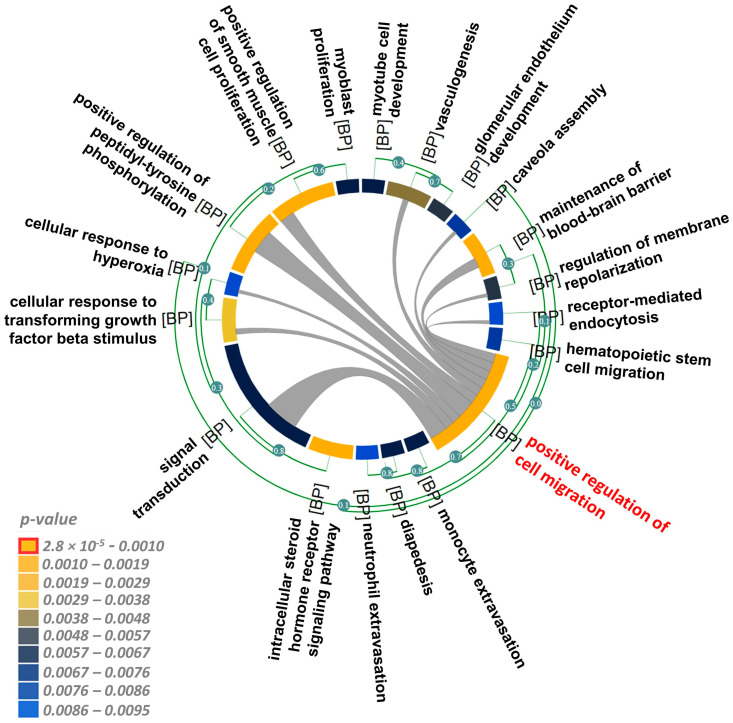
Chord diagram of gene ontology biological process gene set enrichment of genes positively correlated with *NR3C1* expression in breast cancer. The length of the element is proportional to the number of genes related to the GO term. The edges inside the chord diagram between the two elements denote the fact that there are common genes between them. Colours indicate *p*-values according to the scale. Red highlights the most significant biological process enhanced by GR action.

**Table 1 cells-12-00784-t001:** Sample and patient cohort.

Tissue Type	Sample Number	Method	Availability
normal tissues (54 different types *)	184	IHC	https://www.proteinatlas.org/ (accessed on 11 November 2022)
normal breast	459	RNAseq	https://www.proteinatlas.org/ (accessed on 11 November 2022)
different cancer types (17 different types **)	7931	RNAseq	https://www.proteinatlas.org/ (accessed on 11 November 2022)
breast cancer	16	IHC	https://www.proteinatlas.org/ (accessed on 11 November 2022)
breast cancer cell lines	86	RNAseq	https://depmap.org/portal/ (accessed on 11 November 2022)
breast cancer	4421	RNAseq	http://bcgenex.ico.unicancer.fr/ (accessed on 11 November 2022)
breast cancer	11,359	microarray	http://bcgenex.ico.unicancer.fr (accessed on 11 November 2022)

*,**: see details in Figure 1 and Figure 2, respectively.

**Table 2 cells-12-00784-t002:** Functional analysis (gene set enrichment anaysis for biological process terms) of genes that are positively regulated by NR3C1 in breast cancer.

Common GO-BP Terms	Significant Terms	Microarray (*n* = 10,455)		RNAseq (*n* = 4421)	
		*p*-Value	Associated Genes	*p*-Value	Associated Genes
positive regulation of cell migration	GO:0030335	2.8 × 10^−5^	CAV1, CXCL12, IGF1, PECAM1, S1PR1	5.63 × 10^−6^	CAV1, CDH5, CXCL12, DAB2, F10, F2R, FAM107A, FER, HGF, IGF1, KDR, PDGFD, PDGFRA, PECAM1, PPM1F, PRKCA, S1PR1, SEMA3G, SEMA5A, SPRY2, VSIR
positive regulation of smooth muscle cell proliferation	GO:0048661	1.18 × 10^−4^	IGF1, S1PR1, TGFBR2	2.02 × 10^−3^	CALCRL, IGF1, IL6R, PDGFD, S1PR1, TGFBR2, TLR4
intracellular steroid hormone receptor signaling pathway	GO:0030518	1.3 × 10^−4^	NR3C1, PLPP1	3.67 × 10^−3^	NR3C1, NR3C2, PLPP1
positive regulation of peptidyl-tyrosine phosphorylation	GO:0050731	3.9 × 10^−4^	ENPP2, IGF1, PECAM1	3.53 × 10^−4^	ANGPT4, BMP6, ENPP2, FGF7, HGF, IGF1, IL6R, NRP1, PECAM1, RELN
maintenance of blood–brain barrier	GO:0035633	1.11 × 10^−3^	JAM2, PECAM1GO:0010634	2.61 × 10^−6^	CDH5, CLDN5, DMD, JAM2, JAM3, LAMA2, MBP, PECAM1
cellular response to transforming growth factor beta stimulus	GO:0071560	3.47 × 10^−3^	CAV1, NR3C1	3.34 × 10^−5^	ACVRL1, CAV1, FYN, MEF2C, NR3C1, PDE2A, PDE3A, PDGFD, ZFP36L2
vasculogenesis	GO:0001570	3.98 × 10^−3^	CAV1, TGFBR2	8.86 × 10^−6^	CAV1, ENG, HEG1, KDR, MYOCD, QKI, SOX17, TGFBR2, TIE1, TMEM100
glomerular endothelium development	GO:0072011	4.79 × 10^−3^	PECAM1	2.63 × 10^−3^	CD34, PECAM1
regulation of membrane repolarization during action potential	GO:0098903	4.79 × 10^−3^	CAV1	2.63 × 10^−3^	CACNA2D1, CAV1
signal transduction	GO:0007165	6.15 × 10^−3^	CXCL12, IGF1, NR3C1, PECAM1, PLPP1, SPARCL1	9.17 × 10^−3^	AKAP13, DLC1, KANK2, STARD8
myotube cell development	GO:0014904	6.38 × 10^−3^	IGF1	5.15 × 10^−3^	IGF1, NFATC2
monocyte extravasation	GO:0035696	6.38 × 10^−3^	PECAM1	5.15 × 10^−3^	CCR2, PDGFD
diapedesis	GO:0050904	6.38 × 10^−3^	PECAM1	5.15 × 10^−3^	FER, PECAM1
myoblast proliferation	GO:0051450	6.38 × 10^−3^	IGF1	1.04 × 10^−4^	ATOH8, HGF, IGF1
caveola assembly	GO:0070836	7.97 × 10^−3^	CAV1	8.41 × 10^−3^	CAV1, CAV2
hematopoietic stem cell migration to bone marrow	GO:0097241	7.97 × 10^−3^	JAM2	8.41 × 10^−3^	JAM2, JAM3
positive regulation of epithelial–mesenchymal transition involved in endocardial cushion formation	GO:1905007	7.97 × 10^−3^	TGFBR2	8.41 × 10^−3^	ENG, TGFBR2
receptor-mediated endocytosis of virus by host cell	GO:0019065	9.55 × 10^−3^	CAV1	1.13 × 10^−5^	CAV1, CAV2, EPS15, PIKFYVE
cellular response to hyperoxia	GO:0071455	9.55 × 10^−3^	CAV1	4.98 × 10^−4^	CAV1, FAS, FOXO1
neutrophil extravasation	GO:0072672	9.55 × 10^−3^	PECAM1	4.98 × 10^−4^	JAML, PECAM1, PIK3CG

**Table 3 cells-12-00784-t003:** Functional analysis (gene set enrichment analysis for biological process terms) of genes that are negatively regulated by *NR3C1* in breast cancer.

Description	Significant Terms	*p*-Value	Associated Genes	Studies Microarray (*n* = 10,455), RNAseq (*n* = 4421)
sister chromatid cohesion	GO:0007062	1.37 × 10^−3^	STAG3L3	microarray
protein K29-linked ubiquitination	GO:0035519	1.17 × 10^−5^	UBE2S, UBE2T	RNAseq
protein K27-linked ubiquitination	GO:0044314	1.17 × 10^−5^	UBE2S, UBE2T	RNAseq
protein K6-linked ubiquitination	GO:0085020	2.81 × 10^−5^	UBE2S, UBE2T	RNAseq
cell division	GO:0051301	2.36 × 10^−4^	CDCA3, CDT1, SAC3D1, UBE2S	RNAseq
protein K11-linked ubiquitination	GO:0070979	3.13 × 10^−4^	UBE2S, UBE2T	RNAseq
protein K63-linked ubiquitination	GO:0070534	7.24 × 10^−4^	UBE2S, UBE2T	RNAseq
FAD biosynthetic process	GO:0006747	9.13 × 10^−4^	FLAD1	RNAseq
Golgi to transport vesicle transport	GO:0055108	9.13 × 10^−4^	ARF1	RNAseq
synaptic vesicle budding	GO:0070142	9.13 × 10^−4^	ARF1	RNAseq
negative regulation of protein localization to kinetochore	GO:1905341	9.13 × 10^−4^	CDT1	RNAseq
mitotic cleavage furrow ingression	GO:1990386	9.13 × 10^−4^	ARF1	RNAseq
positive regulation of DNA-dependent DNA replication	GO:2000105	9.13 × 10^−4^	CDT1	RNAseq
RNA phosphodiester bond hydrolysis, endonucleolytic	GO:0090502	1.39 × 10^−3^	POP7, RNASEH2A	RNAseq
meiotic cell cycle	GO:0051321	1.72 × 10^−3^	H2AX, PKMYT1	RNAseq
DNA replication preinitiation complex assembly	GO:0071163	1.83 × 10^−3^	CDT1	RNAseq
response to sorbitol	GO:0072708	1.83 × 10^−3^	CDT1	RNAseq
lysosomal membrane organization	GO:0097212	1.83 × 10^−3^	ARF1	RNAseq
positive regulation of sodium ion transmembrane transport	GO:1902307	1.83 × 10^−3^	ARF1	RNAseq
regulation of DNA replication origin binding	GO:1902595	1.83 × 10^−3^	CDT1	RNAseq
positive regulation of late endosome to lysosome transport	GO:1902824	1.83 × 10^−3^	ARF1	RNAseq
regulation of phospholipid metabolic process	GO:1903725	1.83 × 10^−3^	ARF1	RNAseq
double-strand break repair via homologous recombination	GO:0000724	2.20 × 10^−3^	H2AX, RECQL4	RNAseq
regulation of chromosome organization	GO:0033044	2.74 × 10^−3^	CDT1	RNAseq
deactivation of mitotic spindle assembly checkpoint	GO:1902426	2.74 × 10^−3^	CDT1	RNAseq
DNA replication	GO:0006260	3.46 × 10^−3^	RECQL4, RNASEH2A	RNAseq
DNA replication, removal of RNA primer	GO:0043137	3.65 × 10^−3^	RNASEH2A	RNAseq
dendritic spine organization	GO:0097061	3.65 × 10^−3^	ARF1	RNAseq
positive regulation of protein localization to kinetochore	GO:1905342	3.65 × 10^−3^	CDT1	RNAseq
regulation of receptor internalization	GO:0002090	4.56 × 10^−3^	ARF1	RNAseq
riboflavin metabolic process	GO:0006771	4.56 × 10^−3^	FLAD1	RNAseq
regulation of nuclear cell cycle DNA replication	GO:0033262	4.56 × 10^−3^	CDT1	RNAseq
positive regulation of ER to Golgi vesicle-mediated transport	GO:1902953	4.56 × 10^−3^	ARF1	RNAseq
free ubiquitin chain polymerization	GO:0010994	5.47 × 10^−3^	UBE2S	RNAseq
regulation of DNA-dependent DNA replication initiation	GO:0030174	5.47 × 10^−3^	CDT1	RNAseq
regulation of Arp2/3 complex-mediated actin nucleation	GO:0034315	5.47 × 10^−3^	ARF1	RNAseq
kinetochore organization	GO:0051383	5.47 × 10^−3^	CDT1	RNAseq
mitotic cell cycle	GO:0000278	6.22 × 10^−3^	CDT1, PKMYT1	RNAseq
telomeric D-loop disassembly	GO:0061820	7.29 × 10^−3^	RECQL4	RNAseq
protein polyubiquitination	GO:0000209	8.93 × 10^−3^	UBE2S, UBE2T	RNAseq
DNA replication checkpoint signaling	GO:0000076	9.10 × 10^−3^	CDT1	RNAseq
positive regulation of ubiquitin protein ligase activity	GO:1904668	9.10 × 10^−3^	UBE2S	RNAseq

## Data Availability

All data are presented in the manuscript; public databases from which validation data were obtained are indicated in the Methods section and Table 1.

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
