# Peer review of "Context-Dependent Role of Glucocorticoid Receptor Alpha and Beta in Breast Cancer Cell Behaviour"

_cells, 2023, doi:10.3390/cells12050784_

Round 1

Reviewer 1 Report

Authors have evaluated the expression and function of glucocorticoids receptors (GRs) in different breast cancer cell hallmarks and different cell lines. Despite the big effort performed in the present study, they seem not to reach clear conclusions, as stated by the own authors in some parts of the discussion section, this may be due GRs are widely expressed and differences between cell types seem to be marginal and explain controversy in the literature. Further the call the attention to the use of different types of antibodies in the histological detection which is crucial for diagnostic in patients. Nonetheless the well designing of the experiments, the fact that they use of several technical approaches, and the critical and correct presentation and discussion of their own data allow to this reviewer to reach the decision of endorse the paper for being published upon authors address the following issues.

Main points.

1. Authors states in results and discussion sections that gender affects GRs expression, considering that males expressed a significant higher amount of GRs and its function seem to be both dependent and independent of the ligand (as presented by the own results), it could be expected that the appearance of breast cancer would be exacerbated in males that present increased amount of GRs, which is not the case. Even more considering the competence of oestrogen and GRs for the same promoters of several genes involved in proliferation, as commented by authors in the discussion section.  So, authors should experimentally evaluate the effect of testosterone or any other androgen hormone in the expression and/or function of GRs in those cells type/es where the role of GRs has been claimed to be much more evident or relevant according to their own results.

2. Authors barely mentioned the posttranslational modifications of GRs, but their manuscript lacks any experiments in this issue. This part may be crucial for understanding the effect of GRs and their own regulation, such as enhanced activity. These results may allow to authors to perform a better and much more accurate interpretation of their own data. Otherwise, without them the manuscript may be indifferent for readers due to lack of interest.

Format issues.

Discussion section contains some spelling mistakes, as well as some sentences are incomplete; so, it should be revised carefully. Furthermore, this section contains unnecessary information, which would be much more appropriated if it would be allocated in the introduction section, such as the molecular description of the types of GRs and the differences between alpha and beta GRs. The later would help the reader to understand better the WB figures.

Author Response

Responses to Reviewer’s queries

Reviewer 1

„Authors have evaluated the expression and function of glucocorticoids receptors (GRs) in different breast cancer cell hallmarks and different cell lines. Despite the big effort performed in the present study, they seem not to reach clear conclusions, as stated by the own authors in some parts of the discussion section, this may be due GRs are widely expressed and differences between cell types seem to be marginal and explain controversy in the literature. Further the call the attention to the use of different types of antibodies in the histological detection which is crucial for diagnostic in patients. Nonetheless the well designing of the experiments, the fact that they use of several technical approaches, and the critical and correct presentation and discussion of their own data allow to this reviewer to reach the decision of endorse the paper for being published upon authors address the following issues.”

Response: We thank The Reviewer for his/her overall positive opinion about the work and the submitted manuscript.

Main points.

“1. Authors states in results and discussion sections that gender affects GRs expression, considering that males expressed a significant higher amount of GRs and its function seem to be both dependent and independent of the ligand (as presented by the own results), it could be expected that the appearance of breast cancer would be exacerbated in males that present increased amount of GRs, which is not the case. Even more considering the competence of oestrogen and GRs for the same promoters of several genes involved in proliferation, as commented by authors in the discussion section. So, authors should experimentally evaluate the effect of testosterone or any other androgen hormone in the expression and/or function of GRs in those cells type/es where the role of GRs has been claimed to be much more evident or relevant according to their own results.”

Response: We thank The Reviewer for drawing our attention to this point that we would like to clarify. Indeed, in the Results section (Figure 2B in the revised manuscript) we found that indeed NR3C1 expression was higher in male compared to female in normal breast tissues. To elucidate this issue, in the revised version we performed the same comparison related to cancerous breast as well on 1063 female and 12 male breast cancer tissue specimens (please find on novel Figure 3C). In contrast to normal tissue, in breast cancer NR3C1 showed the opposite, its expression was decreased in male compared to female patients, reaching the level of significance (p=0.055). We did not find association between NR3C1 and age groups, similarly to normal breast tissues.

The lower expression of NR3C1 in male breast cancer seems to be in line with the finding that breast cancer in male is most commonly oestrogen positive and it has a good prognosis (Kornegoor et al 2011; Cardoso et al, 2018).

We thank the Reviewer for his/her accurate remark, and we believe, that including the same comparison in normal and cancerous samples clarifies this question and adds a value to the findings which we included into the novel version of the manuscript as well (please find highlighted (lines 335-337) in the Results section, on Figure 3C (line 347) and at the beginning of the Discussion section (lines 467-470).

“2. Authors barely mentioned the posttranslational modifications of GRs, but their manuscript lacks any experiments in this issue. This part may be crucial for understanding the effect of GRs and their own regulation, such as enhanced activity. These results may allow to authors to perform a better and much more accurate interpretation of their own data. Otherwise, without them the manuscript may be indifferent for readers due to lack of interest.”

Response: We agree with the Reviewer that the posttranscriptional modifications of GR have great impact on receptor function. Regarding GR, phosphorylation at least 6 serine residues, sumoylation and acetylation at several specific lysine residues have been reported. Also, all known GR isoforms contain the target lysine residue (at 419 position), therefore the level of all GR isoforms was suggested to be regulated by ubiquitin-mediated degradation (Vandevyver et al. 2014). Additionally, the role of methylation, nitrosylation and oxidation has been also raised in association of GR function. These posttranslational modifications of GR and their functions have been discussed in numerous excellent reviews (Duma et al. 2006; Oakley & Cidlowski 2011; Vandevyver et al. 2014; Noureddine et al. 2021).

Indeed, various posttranslational modifications provide additional receptor heterogeneity for controlling the GR action. As in our experimental design we used the same cell lines and in one experiment only GR expression level or α/β isoform or the presence of the ligand were modulated, we may assume that the experienced differences were mainly due to these parameters and not to different posttranslational modifications. However, we agree with the Reviewer, that these should be further investigated as posttranslational modifications appear during glucocorticoid response.

In our study the main goal was 1) to investigate the possibilities of detection of GR expression in breast cancer in terms of diagnostic perspective, 2) to validate the specificity of a GRβ antibody and 3) to scrutinize the role of α/β isoform expression, the presence of the oestrogen receptor and ligand on breast cancer cell behaviour.

Given that the assessment of the role of even one type of posttranscriptional modification requires a whole study (such as Wilkinson et al. 2018), this clearly exceeds the frame of our goals and study.

Nevertheless, we addressed this issue in the Introduction (legend of Figure 1, line 114-129) and in the Discussion section, please find highlighted (line 613-618).

Format issues.

“Discussion section contains some spelling mistakes, as well as some sentences are incomplete; so, it should be revised carefully.

Response: We thank The Reviewer for his/her accuracy, we performed an extensive language check and corrected spelling mistakes and sentences.

“Furthermore, this section contains unnecessary information, which would be much more appropriated if it would be allocated in the introduction section, such as the molecular description of the types of GRs and the differences between alpha and beta GRs. The later would help the reader to understand better the WB figures”

Response: We are grateful the Reviewer for his/her suggestion. We removed the irrelevant parts from the Discussion section (line 131-149). We also replaced the paragraphs about GR structure and isoforms to the Introduction (please find highlighted, line 93-111). We agree with the Reviewer, this provides a more logical structure and helps the reader in understanding.

We Thank the Reviewer for his/her accuracy and suggestions, we believe that all of them improved the quality of our manuscript.

References:

Cardoso F, Bartlett JMS, Slaets L, van Deurzen CHM, van Leeuwen-Stok E, Porter P, Linderholm B, Hedenfalk I, Schröder C, Martens J, Bayani J, van Asperen C, Murray M, Hudis C, Middleton L, Vermeij J, Punie K, Fraser J, Nowaczyk M, Rubio IT, Aebi S, Kelly C, Ruddy KJ, Winer E, Nilsson C, Lago LD, Korde L, Benstead K, Bogler O, Goulioti T, Peric A, Litière S, Aalders KC, Poncet C, Tryfonidis K, Giordano SH. Characterization of male breast cancer: results of the EORTC 10085/TBCRC/BIG/NABCG International Male Breast Cancer Program. Ann Oncol. 2018 Feb 1;29(2):405-417. doi: 10.1093/annonc/mdx651. PMID: 29092024; PMCID: PMC5834077.

Kornegoor R, Verschuur-Maes AH, Buerger H, Hogenes MC, de Bruin PC, Oudejans JJ, van der Groep P, Hinrichs B, van Diest PJ. Molecular subtyping of male breast cancer by immunohistochemistry. Mod Pathol. 2012 Mar;25(3):398-404. doi: 10.1038/modpathol.2011.174. Epub 2011 Nov 4. PMID: 22056953.

Duma D, Jewell CM, Cidlowski JA. Multiple glucocorticoid receptor isoforms and mechanisms of post-translational modification. J Steroid Biochem Mol Biol. 2006 Dec;102(1-5):11-21. doi: 10.1016/j.jsbmb.2006.09.009. Epub 2006 Oct 27. PMID: 17070034.

Oakley RH, Cidlowski JA. Cellular processing of the glucocorticoid receptor gene and protein: new mechanisms for generating tissue-specific actions of glucocorticoids. J Biol Chem. 2011 Feb 4;286(5):3177-84. doi: 10.1074/jbc.R110.179325. Epub 2010 Dec 13. PMID: 21149445; PMCID: PMC3030321.

Vandevyver S, Dejager L, Libert C. Comprehensive overview of the structure and regulation of the glucocorticoid receptor. Endocr Rev. 2014 Aug;35(4):671-93. doi: 10.1210/er.2014-1010. Epub 2014 Jun 17. PMID: 24937701.

Noureddine LM, Trédan O, Hussein N, Badran B, Le Romancer M, Poulard C. Glucocorticoid Receptor: A Multifaceted Actor in Breast Cancer. Int J Mol Sci. 2021 Apr 24;22(9):4446. doi: 10.3390/ijms22094446. PMID: 33923160; PMCID: PMC8123001.

Wilkinson L, Verhoog N, Louw A. Novel role for receptor dimerization in post-translational processing and turnover of the GRα. Sci Rep. 2018 Sep 24;8(1):14266. doi: 10.1038/s41598-018-32440-z. PMID: 30250038; PMCID: PMC6155283.

Reviewer 2 Report

Authors will do the following:

1. Improve the English editing.

2. Please provide the mechanism of action of glucocorticoid receptor alpha and beta in breast cancer cell, graphical view.

3. In Figure 3 is not clearly provide sound to each others. Please mark in the figures how you will differentiate to controll vs LumA vs TNBC.

4. Figure 4 also will do the like point 3.

5. Improve the Discussion as well as conclusion

Author Response

Responses to Reviewer’s queries

Reviewer 2

„Authors will do the following:

  1. Improve the English editing.”

Response: We thank the Reviewer for his/her accuracy, we performed an extensive language check and corrected spelling mistakes and sentences.

„2. Please provide the mechanism of action of glucocorticoid receptor alpha and beta in breast cancer cell, graphical view.”

Response: As per the Reviewer’s request we included a novel Figure presenting the mechanism of action of glucocorticoid receptor alpha and beta in breast cancer cell. Please find Figure 1 in the revised manuscript.

„3. In Figure 3 is not clearly provide sound to each others. Please mark in the figures how you will differentiate to controll vs LumA vs TNBC.”

Response: Breast cancer subtypes were determined by extensive pathological evaluation as part of the routine diagnostic procedure before GR immunostaining. Following classification by oestrogen (ER), progesterone receptor (PR), Her2 and Ki-67 staining samples were categorized as control, luminal A and triple-negative breast cancer (TNBC) by the pathologist (E. Tóth) according to Tsang et al. (Tsang & Tse. Molecular Classification of Breast Cancer. Adv Anat Pathol. 2020;27(1):27-35.) (Luminal A samples were defined by strong estrogen receptor staining combined with low (<20%) Ki-67 indices. TNBC were selected based on missing ER, PR and Her2 staining beside positive controls.) After classification and routine pathological diagnostics samples were provided for further GR immunohistochemical study. Control samples were selected from an FFPE block of a surgical specimen of a ER+ breast cancer patient where no malignant tissue was identified by the pathologist or adjacent parts to tumours. We indicated these in the methods section, please find it highlighted (lines 162-167).

„4. Figure 4 also will do the like point 3.”

Response: On Figure 4 (Figure 5 in the revised version) is a representative image, on which all three tumors (A, B and C) are triple negative subtype. However, we find similar pattern in ER+ tumors as well, that we indicated in the Results as:

“Staining was heterogeneous among samples irrespective of tumour type. Both GRα and GRβ exhibited mostly cytoplasmic localization in tumours, in some samples with nuclear positivity” Please find highlighted (lines 356-358).

Subtype classification was done as described in the response for query 3.

“5. Improve the Discussion as well as conclusion”

Response: Paying attention to all three Reviewers’ query we improved the Discussion and Conclusion section, replacing irrelevant parts into the Introduction and including novel parts as per the queries. The Discussion now is organized according to the aims of the study for clarity.

Finally, we would like to thank the Reviewer for his/her accurate remarks and suggestions that improved the manuscript.

Reviewer 3 Report

Glucocorticoid receptors have been studied in multiple contexts but the results have mostly been heterogenous due to different confirmations and use of different antibodies. Here in the current manuscript, the authors solve this conendrum by addressing this question with multi-arm scientific approach. Using immunohistochemistry and sequencing data, the authors show the presence of differential expression of alpha and beta confirmations of the gluco-corticoid receptors in different cancer cell lines. They also connected this with the expression of Oestrogen receptor that differentially regulates the expression of GRs. The authors nicely draw the conclusions that the expression of GRa or GRb depends on the expression of OR or ER context and that regulates its functions in cell migration or cancer invasiness. The overall pattern of manuscript is nice and easy to follow. The figures are neat and the western blots have full blots which makes it easy to understand. 

I would recommend the publication of this manuscript with a small improvement that can could be added in the discussion session.

In the current manuscript the authors nicely demonstrate one context that is ER that regulates different expression patterns of GRa and GRb. How about other contexts like growth factor receptors or the cascades in which ER is itself regulated by other receptors which in turn could be regulating GR expression. Also, if there is any feedback loop that GR could back regulate ER expression in cells. These are interesting phenomena and would require further investigations.

Author Response

Responses to Reviewer’s queries

Reviewer 3

„Glucocorticoid receptors have been studied in multiple contexts but the results have mostly been heterogenous due to different confirmations and use of different antibodies. Here in the current manuscript, the authors solve this conendrum by addressing this question with multi-arm scientific approach. Using immunohistochemistry and sequencing data, the authors show the presence of differential expression of alpha and beta confirmations of the glucocorticoid receptors in different cancer cell lines. They also connected this with the expression of Oestrogen receptor that differentially regulates the expression of GRs. The authors nicely draw the conclusions that the expression of GRa or GRb depends on the expression of OR or ER context and that regulates its functions in cell migration or cancer invasiness. The overall pattern of manuscript is nice and easy to follow. The figures are neat and the western blots have full blots which makes it easy to understand.

I would recommend the publication of this manuscript with a small improvement that can could be added in the discussion session.

Response: We thank The Reviewer for his/her overall positive opinion of the manuscript.

“In the current manuscript the authors nicely demonstrate one context that is ER that regulates different expression patterns of GRa and GRb. How about other contexts like growth factor receptors or the cascades in which ER is itself regulated by other receptors which in turn could be regulating GR expression. Also, if there is any feedback loop that GR could back regulate ER expression in cells. These are interesting phenomena and would require further investigations.”

Response: As per the Reviewer’s request we addressed the abovementioned queries in the Discussion section: i.) other contexts like growth factor receptors; ii.) the cascades in which ER is itself regulated by other receptors which in turn could be regulating GR expression and iii.) feedback loops that GR could back-regulate ER expression in cells.

Please find the added paragraphs below and highlighted in the supplemented discussion.

i.) GR action in other contexts like growth factor receptors:

“Besides interactions between nuclear receptors, crosstalk between GR and growth factors has been also reported [47,48]. EGFR, one of the most active growth factors exerting strong growth-promoting effects in mammary epithelium [47] can interact with GR through both genomic and epigenomic processes [48]. Regarding crosstalk with other growth factors, GR was shown to be a required effector of TGFβ1-induced p38 MAPK signaling [49] and it suppressed the transcription of the insulin receptor substrate 1 (IRS-1), which mediates insulin-like growth factor (IGF) signals [20].” (line 535-541)

ii.) the cascades in which ER is itself regulated by other receptors which in turn could be regulating GR expression:

“In the complex interaction network of GR-ER, there are indirect processes in which ER is itself regulated by other receptors which in turn could be regulating GR expression. Signal transduction by Her2 and epidermal growth factor (EGFR) receptors was described to alter the phosphorylation of ER and ER-dependent signaling irrespective of the presence of ER ligands [45]. In addition, both oestrogen and growth factor signaling pathways regulate the secretion of vascular endothelial growth factors that stimulate tumour-associated an-giogenesis [45]. Also, evidence suggested that cross-talk between ER and growth factor re-ceptor pathways contributed to the development of tamoxifen resistance in breast cancer. Signaling via the EGFR and Her2 could activate both ER and the ER coacti-vator AIB1. In turn, ER located in the cell membrane can activate the growth factor receptor pathways [46].” (line 524-534)

iii.) feedback loops that GR could back-regulate ER expression in cells

“The GR-ER crosstalk is additionally illustrated by feedback loops where GR could back-regulate ER expression. While there is no GRE identified in the promoter of ER, indirect regulation of ER by GR can be hypothesized as feedback loop control. The promoters of ERα contain multiple predicted and validated transcription factors-binding sites [42,43]. Several of them are in indirect interaction with GR such as ER itself, BRCA1, ZEB2, NF-κB, or circadian genes [42,43]. In addition, DNMT1 and ZEB1, as GR-regulated genes, can induce ERα promoter methylation, hence down-regulation of ERα expression [42]. Similarly, histone acetylation and methylation also play a role in ER expression, while histone acetyltransferases and demethylase are also regulated by GR at the promoter level [43]. Additionally, another way in which GR and ER signaling interact is by decreasing levels of free oestrogen through GR-mediated activation of oestrogen sulfotransferase [44].” (line 512-523)

We thank the Reviewer for his/her accurate remarks and suggestions that improved the revised manuscript.

Round 2

Reviewer 1 Report

Authors have addressed my concerns, so I endorse its publication in the jou

Author Response

Comments and Suggestions for Authors

“Authors have addressed my concerns, so I endorse its publication in the journal”

Response: We thank The Reviewer for his/her accurate work and all the suggestions that improved the quality of our manuscript.